

# Learning to predict pain: differences in people with persistent neck pain and pain-free controls

Daniel S. Harvie[1,2], Jeroen D. Weermeijer[3], Nick A. Olthof[1,2] and Ann Meulders[4,5]

[1] The Hopkins Centre, Menzies Health Institute Queensland, Griffith University, Gold Coast, QLD, Australia
[2] School of Allied Health Sciences, Griffith University, Gold Coast, QLD, Australia
[3] Center for Contextual Psychiatry, KU Leuven, Leuven, Belgium
[4] Research Group Health Psychology, KU Leuven, Leuven, Belgium
[5] Experimental Health Psychology, Maastricht University, Maastricht, The Netherlands

Corresponding author
Daniel S. Harvie,
d.harvie@griffith.edu.au

## ABSTRACT

**Background:** Learning to predict threatening events enables an organism to engage in protective behavior and prevent harm. Failure to differentiate between cues that truly predict danger and those that do not, however, may lead to indiscriminate fear and avoidance behaviors, which in turn may contribute to disability in people with persistent pain. We aimed to test whether people with persistent neck pain exhibit contingency learning deficits in predicting pain relative to pain-free, gender-and age-matched controls.

**Method:** We developed a differential predictive learning task with a neck pain-relevant scenario. During the acquisition phase, images displaying two distinct neck positions were presented and participants were asked to predict whether these neck positions would lead to pain in a fictive patient with persistent neck pain (see fictive patient scenario details in Appendix A). After participants gave their pain-expectancy judgment in the hypothetical scenario, the verbal outcome (PAIN or NO PAIN) was shown on the screen. One image (CS+) was followed by the outcome "PAIN", while another image (CS−) was followed by the outcome "NO PAIN". During the generalization phase, novel but related images depicting neck positions along a continuum between the CS+ and CS− images (generalization stimuli; GSs) were introduced to assess the generalization of acquired predictive learning to the novel images; the GSs were always followed by the verbal outcome "NOTES UNREADABLE" to prevent extinction learning. Finally, an extinction phase was included in which all images were followed by "NO PAIN" assessing the persistence of pain-expectancy judgments following disconfirming information.

**Results:** Differential pain-expectancy learning was reduced in people with neck pain relative to controls, resulting from patients giving significantly lower pain-expectancy judgments for the CS+, and significantly higher pain-expectancy judgments for the CS−. People with neck pain also demonstrated flatter generalization gradients relative to controls. No differences in extinction were noted.

**Discussion:** The results support the hypothesis that people with persistent neck pain exhibit reduced differential pain-expectancy learning and flatter generalization gradients, reflecting deficits in predictive learning. Contrary to our hypothesis,

no differences in extinction were found. These findings may be relevant to understanding behavioral aspects of chronic pain.

## INTRODUCTION

Learning to predict threatening events is highly adaptive as it enables an organism to respond in anticipation with protective action to avoid or limit injury (*Enquist, Lind & Ghirlanda, 2016*). A prime mechanism underlying the prediction of threatening events is associative learning, and more specifically classical conditioning (*Pavlov, 1927*). The outcome of an aversive conditioning or fear conditioning procedure is apparent when an initially inert stimulus (conditioned stimulus; CS) such as a neutral movement, comes to elicit defensive behavioral responses (conditioned response; CR) such as increased autonomic arousal, verbal fear reports, and avoidance behavior, after contingent pairing with an aversive stimulus such as pain (unconditioned stimulus; US) that innately elicits a response (unconditioned response).

Contemporary biopsychosocial models of (musculoskeletal) chronic pain consider pain-related fear and avoidance as key contributors in the transition from acute to chronic pain (*Meulders, 2019*; *Vlaeyen, 2015*). Previous research indeed revealed that high scores on pain-related fear and avoidance beliefs questionnaires are associated with poor outcomes following injury and increased disability in a range of conditions such as neck pain (*Landers et al., 2008*; *Nederhand et al., 2004*) and back pain (*Crombez et al., 1999*; for review see *Vlaeyen & Linton, 2012*). These data suggest that it is not merely the severity of the injury and the associated pain intensity, but rather an individual's behavioral response to pain that paves the way to chronification. Given that injury-related learning shapes these individual responses to pain, learning processes may also be involved in determining functional outcomes. Following this reasoning, aberrant conditioning responses in people with persistent pain may exist when compared to people without persistent pain, which in turn may contribute to their vulnerability (*Meulders, 2019*).

Learning to predict threatening events is highly adaptive—a critical component of adaptive pain-related fear conditioning is learning to respond *selectively* to predictors of harm thus equally important is safety learning, which is, learning to refrain from responding to cues that do not predict harm. Failure to differentiate between danger and safety cues appears to be characteristic of pathology in generalized anxiety disorder (*Duits et al., 2015*; *Grillon, 2002*), post-traumatic stress disorder (*Jovanovic et al., 2012*; *Lissek et al., 2005*) schizophrenia (*Clifton et al., 2017*), and obsessive compulsive disorder (*Apergis-Schoute et al., 2017*) where affective and behavioral defensive responses are unduly triggered.

Furthermore, rapid adaptation to a dynamic environment requires balancing discrimination and generalization towards novel situations and stimuli. *Stimulus generalization*

(*Honig & Urcuioli, 1981*; *Kalish, 1969*) allows extrapolating the acquired value of one stimulus to novel stimuli (generalization stimuli; GSs) based on (non-)perceptual similarities and has the advantage that defensive action can be taken in the absence of prior experience or new learning. Excessive fear generalization to safe situations however may induce persistent anticipatory anxiety and excessive avoidance behavior initiating a pathway towards disability (*Dymond et al., 2015*; *Meulders et al., 2014*, *2017*).

In recent years, associative learning deficits have been reported in several chronic pain populations, including fibromyalgia (*Jenewein et al., 2013*; *Meulders, Jans & Vlaeyen, 2015*; *Meulders et al., 2017*), chronic unilateral hand pain (*Meulders et al., 2014*), and chronic back pain (*Klinger et al., 2010*; *Schneider, Palomba & Flor, 2004*; for review see *Harvie et al., 2017*). In line with the anxiety literature, current evidence suggests that people with persistent pain do not show a tendency to exhibit exaggerated CRs to pain-associated cues (CS+) but rather impaired safety learning (i.e., greater pain-expectancy and fear in response to safety cues (CS−)) as well as "overgeneralization" (i.e., greater pain-expectancy and fear in response to novel stimuli that share features with the pain-associated cues). Impaired predictive learning and stimulus generalization may not only become maladaptive when it spreads excessively to safe stimuli, but also when it persists despite disconfirmation of the expected outcome. Indeed previous studies suggests that some patients may show resistance to extinction of differential conditioning (*Schneider, Palomba & Flor, 2004*) or generalization (*Meulders et al., 2017*) following corrective feedback, which may further maintain disability in chronic pain conditions.

Second only to back pain, neck pain is the greatest cause of years lived with disability worldwide (*Vos et al., 2015*). Yet, it is unknown whether associative learning deficits are a feature of all or only specific persistent pain conditions, and thus it is unclear whether they are prevalent among people with persistent neck pain. To date, it is commonly accepted that the conditions for learning a causal relationship between two neutral events closely resemble those that underlie classical conditioning with a biologically significant US (*Dickinson, 1980*; *Shanks & Dickinson, 1988*). As such, *contingency learning* (*De Houwer & Beckers, 2002*) or predictive learning tasks, in which participants are required to give outcome-expectancy judgments when presented with certain cues, offer a valid proxy to tap into the basic processes underlying classical fear conditioning (*Boddez et al., 2013*). Therefore, in line with previous studies using contingency learning tasks in chronic pain populations (*Meulders et al., 2014*, *2017*; *Meulders, Jans & Vlaeyen, 2015*), we developed a predictive learning task based around a clinical neck pain scenario to investigate whether people with persistent neck pain show deficits in predictive learning relative to pain-free, age- and gender-matched controls. We hypothesized that people with persistent neck pain would show (1) less differential predictive learning at the end of acquisition (i.e., a reduced difference in CS+ vs. CS− pain-expectancy judgments), (2) overgeneralization of pain-expectancy (e.g., flatter generalization gradients and higher pain-expectancy judgments for GSs more similar to the CS−) and (3) delayed extinction of acquired differential pain-expectancies (i.e., persistent differential CS+ vs. CS− pain-expectancy) relative to healthy controls.

## MATERIALS AND METHODS

### Participants

Thirty individuals with persistent neck pain (14 males, mean age = 50.9 years, SD age = 15.6) with diagnosis confirmed by a qualified physiotherapist, and 30 age-and gender-matched pain-free controls (14 males, mean age = 49.9 years, SD age = 14.6) were recruited; the target sample size of 60 was a priori determined based on previous literature. That is, similar studies in other pain groups have reported moderate effects ($\eta_p^2 = 0.07$) for chronic pain vs. healthy control differences in pain-expectancy learning (*Meulders, Jans & Vlaeyen, 2015*). Using this effect size, we calculated that 30 subjects would be needed for 80% power to detect an interaction between stimulus type and group. To assist in accounting for multiple analyses we doubled the calculated sample size. Other studies have found significant between group effects for similar outcomes with just 48 subjects, providing further sample size support (*Meulders et al., 2014*). On average, included participants had moderate pain (mean pain intensity = 4.5, SD = 1.2 on a 11-point scale) and reported moderate disability (mean disability = 30%, SD = 12%) as assessed by the Neck Disability Index (NDI), where 8–29% = mild, 30–39% = moderate and above 40% = severe disability (*Vernon & Mior, 1991*). Participation was voluntary and no monetary incentive was provided. Participants with persistent neck pain were required to have had neck pain for a period of at least 3 months, whilst control participants were required to have no neck or other pain, and to have no history of pain lasting longer than 3 months. Participants in both groups were excluded on the basis of neurological deficits (i.e., stroke, dementia, Alzheimer, cervical radiculopathy, cervical myelopathy), diagnosed psychological disorders (i.e., PTSD, Schizophrenia, Bi-polar Disorder, Dissociative Disorder, Panic Disorder), serious orthopaedic deficit that may indicate ongoing physical driver of pain and disability (i.e., fracture, spondylolisthesis), or were currently taking certain medications that might alter cognitive functioning (i.e., opioids, neuroleptics and anxiolytics). All participants were further required to have normal or corrected to normal vision. Participants with neck pain were recruited through the Recover Injury Research Centre database (Gold Coast, Australia), local clinics (Gold Coast, Australia), word-of-mouth, and social media. Patients were asked to nominate a same-aged (+/− 5 years) friend or family member of the same gender to volunteer as a control subject. Where participants were not able to bring an age-and gender-matched control, participants were supplemented through recruitment posters placed around the university campus, word-of-mouth, and social media. Initial contact and eligibility screening was undertaken by telephone. Each participant provided written informed consent before the start of the experiment. The protocol was approved by the Griffith University institutional Human Research Ethics Committee (GFU Reference number: 2017/710). The protocol and analysis plan were not preregistered. After the experiment, we assessed the following psychological traits using questionnaires for descriptive purposes: (1) positive and negative affect (Positive and Negative Affect Schedule; PANAS) (*Watson, Clark & Tellegen, 1988*), (2) trait anxiety (State-Trait Anxiety Inventory; STAI)

Table 1 Demographics and participant characteristics for both the persistent neck pain patients and pain-free controls.

| | Persistent neck pain (n = 30) | | | Pain-free controls (n = 30) | | | Mean diff. (95% CI) |
|---|---|---|---|---|---|---|---|
| | Mean | SD | Range | Mean | SD | Range | |
| Age (in years) | 50.90 | 15.87 | 18–72 | 49.90 | 14.55 | 23–73 | −1.00 [−0.34 to −1.66] |
| PANAS–positive affect | 31.60 | 8.06 | 19–48 | 30.73 | 7.68 | 14–47 | −0.87 [1.94 to −3.67] |
| PANAS–negative affect | 12.50 | 3.18 | 10–21 | 12.53 | 3.57 | 10–23 | 0.03 [1.26 to −1.20] |
| STAI-T–total | 38.87 | 10.65 | 21–58 | 36.60 | 10.89 | 20–63 | −2.27 [1.68 to −6.22] |
| PCS–total | 13.93 | 10.04 | 2–36 | 7.97 | 7.03 | 0–32 | −5.97 [−2.81 to −9.12] |
| PCS–rumination | 4.3 | 3.54 | 0–14 | 3.17 | 3.13 | 0–12 | −1.13 [0.12 to −2.39] |
| PCS–magnification* | 3.07 | 3.04 | 0–10 | 1.63 | 1.92 | 0–8 | −1.43 [−0.48 to −2.38] |
| PCS–helplessness** | 6.57 | 4.15 | 1–15 | 3.17 | 2.74 | 0–12 | −3.40 [−2.16 to −4.64] |
| PHQ-9*–total | 7.47 | 5.64 | 0–18 | 3.9 | 4.49 | 0–20 | −5.97 [−2.81 to −9.12] |
| TSK-11–total** | 24.10 | 5.49 | 12–36 | 18.97 | 5.18 | 11–28 | −5.13 [−3.24 to −7.03] |
| TSK-11–somatic focus | 10.77 | 3.03 | 5–17 | 8.63 | 2.86 | 5–14 | −2.13 [−1.04 to −3.23] |
| TSK-11–avoidance | 13.33 | 3.25 | 7–23 | 10.33 | 2.60 | 6–15 | −3.00 [−1.98 to −4.02] |
| NDI–total | 30.0% | 12.4 | 12–54% | – | – | – | |
| Current pain (0–10) | 4.07 | 2.21 | 0–8 | 0.1 | 0.31 | 0–1 | −3.97 [−3.38 to −4.55] |
| Av. pain last 2 weeks (0–10) | 4.47 | 1.18 | 0–7 | 0.13 | 0.43 | 0–2 | −4.33 [−3.84 to −4.83] |
| Worst pain 2 weeks (0–10) | 6.8 | 2.53 | 0–10 | 0.57 | 1.14 | 0–7 | −6.27 [−5.52 to −7.01] |

Notes:
* $p < 0.05$.
** $p < 0.01$.
PANAS, positive and negative affect schedule; STAI-T, trait version of the state-trait anxiety inventory; PCS, pain pain-catastrophizing scale; PHQ-9, patient health questionnaire; TSK-11, tampa scale for kinesiophobia; NDI, neck disability index; Range, the minimum and maximum value for the given variable.

(*Spielberger et al., 1983*), (3) pain catastrophizing (Pain Catastrophizing Scale; PCS) (*Sullivan, Bishop & Pivik, 1995*), (4) fear of movement (Tampa Scale for Kinesiophobia; TSK-11) (*Woby et al., 2005*) and general health (Patient Health Questionnaire; PHQ-9) (*Kroenke, Spitzer & Williams, 2001*). Participants also rated their current pain, habitual, and highest pain experienced in the last two weeks. A detailed overview of the sample characteristics can be found in Table 1.

## Software and stimulus material

This study was programmed using Affect 4.0, a Windows-based freely available experimental software package (*Spruyt et al., 2010*). The cues that served as conditioned stimuli (CS+ and CS−), and the generalization stimuli (GS1, GS2, GS3, GS4 and GS5) consisted of images of an avatar displaying different neck positions (Fig. 1). The images were created with the free 3-D animation software, DAZ studio 4.9 pro (https://www. daz3d.com). The CS+ and CS− images were set at 75° of head rotation in opposing directions. The GS1-5 images set at 25° increments from 50° left rotation to 50° right rotation, starting in the direction of the CS+. All stimuli were presented on a 17-inch computer monitor, at the center of the screen on a black background. The outcome during acquisition was one of two verbal labels displayed on the computer screen, in font Arial with font size 48: "PAIN" and "NO PAIN", during the generalization phase the following outcome was presented "NOTES UNREADABLE".

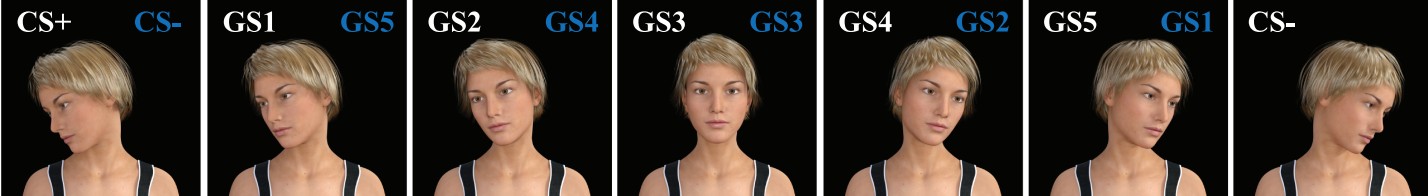

**Figure 1** Conditioned stimuli (CS+ and CS−) and generalization stimuli (GS1-5) presented with CSs on the extremes and the intermediate GSs in between representing the generalization gradient. Which stimulus served as the CS+ and the CS− (i.e., white vs. blue labels) was counter-balanced across participants.

## Procedure

Data collection was undertaken in a laboratory environment, free from noise and visual distraction. The task was undertaken while seated in front of a laptop with a 17″ screen, placed on a table against a blank wall. The experimenter remained behind a curtain out of view. On arrival, written information was provided, and participants signed informed consent. Upon starting the computerized task, participants read a set of on-screen standardized instructions (Supplemental File 1), during which they could ask for clarification. Once the instructions had been read and questions answered, the computerized task was commenced. The experimenter (JDW or NO) would remain inside the room but out of view. The entire experimental session took approximately 30 min including questionnaires.

## Neck pain scenario predictive learning task

Based on previous work (*Meulders et al., 2014*), we designed a predictive learning task based around a clinical neck pain scenario (Fig. 2). During the task, participants were asked to predict whether certain neck positions would lead to pain in a fictive chronic neck pain patient, rather than in themselves. At the start of each trial, an avatar displaying a neck position was presented. After 2 s, the question *T. what extent do you expect Alex to experience pain in this position?* appeared above the image. Additionally, an 11-point numerical rating scale (NRS), ranging from 0 "not at all" to 10 "very much", was presented horizontally at the top of the screen, below the presented question. Participants answered this question by clicking on a value with the left mouse button; their choice was visualized by a white dot appearing on the selected value of the NRS. Participants could then confirm their answer by pressing the spacebar. After answer confirmation, the image would disappear, followed by one of the three possible outcomes: "PAIN", "NO PAIN", or "NOTES UNREADABLE" for 1.5 s, depending on whether the neck position was assigned as the CS+, CS−, or GS respectively. Each trial was followed by a 1 s interval before onset of the next trial. The pain-expectancy judgments served as primary outcome for the evaluation of predictive learning and stimulus generalization.

The predictive learning task consisted of four phases: a familiarization phase, an acquisition phase, a generalization phase, and an extinction phase (Table 2). The familiarization phase comprised one block, containing three trials. There was one CS+ presentation, one CS− presentation, and one GS3 presentation. Stimuli were presented in

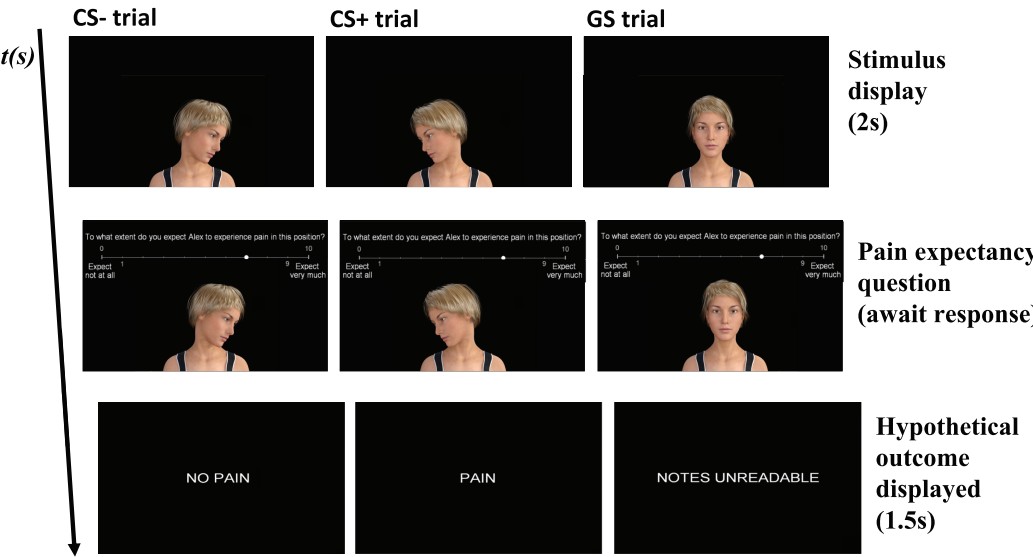

**Figure 2** **Flow chart of the predictive learning task, with a CS−, CS+ and GS trial as examples.** Middle row displays the question: *To what extent do you expect Alex to experience pain in this position?* CS+, conditioned stimulus paired with "PAIN"; CS−, conditioned stimulus paired with "NO PAIN"; GS, generalization stimulus paired with "NOTES UNREADABLE".

**Table 2 Experimental design.**

| Familiarization | Acquisition* | Generalization | Extinction |
|---|---|---|---|
| $\begin{pmatrix} 1 \times CS+ \\ 1 \times GS3 \\ 1 \times CS- \end{pmatrix}$ | $4 \times \begin{pmatrix} 4 \times CS+ \\ 4 \times CS- \end{pmatrix}$ | $2 \times \begin{pmatrix} 1 \times CS+ \\ 1 \times GS1\text{-}5 \\ 1 \times CS- \end{pmatrix}$ | $5 \times \begin{pmatrix} 4 \times CS+ \\ 4 \times CS- \end{pmatrix}$ |

**Notes:**
* 75% reinforcement rate of CS+.
CS+, conditioned stimulus paired with "PAIN"; CS−, conditioned stimulus paired with "NO PAIN"; GS, generalization stimulus. Stimuli were presented in semi-random fashion within each block (denoted by brackets) with the condition that no stimulus would be presented more than twice consecutively.

randomized order. During the familiarization phase, participants were allowed to ask for clarification if needed. The acquisition phase comprised four blocks, each containing eight trials. Each acquisition block included four CS+ presentations and four CS− presentations. Stimuli were presented in semi-randomized order, ensuring that no more than two consecutive trials were of the same stimulus type. The CS+ was followed by the "PAIN" outcome in 75% of the trials, and by the "NO PAIN" outcome in the other 25% of the trials. The CS- was always followed by the "NO PAIN" outcome. Which stimulus served as the CS+ and CS− was counterbalanced between participants. The generalization phase included two blocks, each containing seven trials. Each generalization block consisted of one CS+ presentation, one CS− presentation and one presentation of each GS. Stimuli were presented randomized within each block. The CS+ was again followed by the "PAIN" outcome, whereas the CS− was followed by the "NO

PAIN" outcome. The GS trials were followed by the outcome "NOTES UNREADABLE" in order to prevent extinction. The extinction phase comprised five blocks, each containing eight trials. Each extinction block included four CS+ presentations and four CS− presentations. Stimuli were presented in semi-randomized order. During the extinction phase, both the CS+ and the CS− were followed by the "NO PAIN" outcome.

## Data analysis overview

To test our first hypothesis as to whether people with persistent neck pain show less differential pain-expectancy judgments relative to pain-free controls, we conducted a $2 \times 2 \times 4$ Repeated Measures (RM) Analysis of Variance (ANOVA) with Group (patient/control) as between-subjects factor and Stimulus type (CS+/CS−) and Block (ACQ1-4) as within-subjects factors. Planned between-group comparisons were then carried out to examine group differences in CS+ and/or CS− responses. This analysis enabled us to first verify whether acquisition of differential conditioning occurred, this would be supported by a Stimulus Type × Block interaction. Evidence supporting that the neck pain group showed less differential pain-expectancy learning, would be provided by a significant interaction between Group and Stimulus Type (i.e., less differential learning) and/or an interaction between Group, Stimulus Type and Block (i.e., delayed differential learning).

To test our second hypothesis that people with persistent neck pain would show overgeneralization relative to pain-free controls, we conducted a $2 \times 2 \times 7$ RM ANOVA with Group (patient/control) as between-subjects factor, and Stimulus type (CS+/GS1-5/CS−) and Block (GEN1-2) as within-subjects factors. Additionally, linear trend analyses were carried out to further investigate differences in stimulus generalization. To enable this, an index of the linear slope of generalisation was calculated for each participant CS+/GS1-5/CS− across the generalisation phase, using the Microsoft excel linear slope function. The slopes were then compared using an independent samples $t$-test. CS+ expectancy ratings were then compared to GS and CS− expectancy ratings within each group, to determine whether the CS expectancy ratings were distinct to other stimuli. Finally, these contrasts were repeated for the CS− expectancy ratings relative to the GS and CS expectancy ratings to further probe differential responding and generalization. Holm-Bonferroni corrections were applied to account for multiple comparisons (*Abdi, 2010*). Statistical support for overgeneralization in the neck pain group would firstly require observing a significant interaction between Group and Stimulus Type in the generalization phase. Secondly, observing a statistically flatter linear gradient of CS+/GS1-5/CS− pain-expectancy judgments in the patient group, representing higher ratings on the safe side of the gradient compared with the healthy controls. Finally, follow-up comparisons specifically between CS− pain-expectancy and GS pain-expectancies within each group would further confirm non-differential responding in the patient group. Specifically, we expect that more GS pain-expectancies will differ from CS− expectancies in the control group relative to the patient group. The CS− vs. GS contrasts were emphasized because the generalization decrement frequently renders CS+

expectancies different to all other stimuli, making it less sensitive to detect between-group differences.

To test our third hypothesis as to whether people with persistent neck pain show impaired extinction as compared to pain-free controls, we conducted a $2 \times 2 \times 6$ RM ANOVA with Group (patient/control) as between-subjects factor and Stimulus Type (CS+/CS−) and Block (ACQ4, EXT1-5) as within-subjects factors. Planned within-group comparisons comparing the CS+ to the CS− at the end of extinction as compared to the end of acquisition were carried out to confirm successful extinction within each group. To further evaluate group-based differences during extinction, we conducted planned between-group comparisons. The statistical criterion for delayed extinction was a significant Group $\times$ Stimulus Type $\times$ Block interaction, which would be indicative of a delayed elimination of the CS+/CS− differential expectancy ratings in the patient group.

For the conducted series of RM ANOVAs; Mauchly's test of sphericity was used to check whether the assumption of sphericity was violated. The Greenhouse-Geisser epsilon correction was reported together with corrected degrees of freedom and corrected $p$-values when this occurred. Partial eta squared was reported for RM ANOVA as a measure of effect size. Planned comparisons were corrected using the Holm-Bonferroni method (*Holm, 1979*) in case of multiple testing. Cohen's $d$ was reported as an index of effect for pairwise comparisons. Both JDW and DH conducted all statistical analyses, independently, with no significant conflict. One immaterial conflict arose due to differences in methods to calculate and compare the linear slopes. Analyses were carried out with Statistica 13.0 (*Statistica TIBCO, 2017*) and JASP 0.9 (*JASP, 2018*).

## RESULTS

*Hypothesis 1: Do people with persistent neck pain show less differential acquisition in pain-expectancy judgments than pain-free controls?*

The analysis revealed a main effect of Stimulus Type, $F_{(1, 58)} = 158.95$, $p < 0.001$, $\eta_p^2 = 0.66$, suggesting that for all participants pain-expectancy judgments for the CS+ were higher than for the CS−. As expected, this difference significantly increased over time, Stimulus Type $\times$ Block interaction, $F_{(3, 174)} = 3.55$, $p < 0.05$, $\eta_p^2 = 0.06$, $\varepsilon = 0.84$ (see Fig. 3). Differential pain-expectancy judgments were significantly smaller in the patient group, $\Delta$(CS+, CS−) controls = 6.05 (SD = 2.67) and patients = 2.68 (SD = 2.69), Stimulus Type $\times$ Group interaction, $F_{(1, 58)} = 23.74$, $p < 0.001$, $\eta_p^2 = 0.10$. This effect was not modulated by time, Stimulus Type $\times$ Block $\times$ Group interaction, $F_{(3, 174)} = 0.13$, $p = 0.92$, $\eta_p^2 = 0.00$.

Analysis across acquisition revealed that patients displayed significantly lower CS+ pain-expectancies, $t_{(58)} = 3.86$, $p < 0.001$, Cohen's $d = 1.00$, $M_{\text{patients}}$ (SD) = 5.11 (2.30) vs. $M_{\text{controls}}$ (SD) = 7.18 (1.83), and significantly higher CS− pain-expectancies, $t_{(58)} = 2.48$, $p = 0.016$, Cohen's $d = 0.64$, $M_{\text{patients}}$ (SD) = 2.43 (2.53) vs. $M_{\text{controls}}$ (SD) =1.13 (1.35).

As expected, these findings show that people with persistent neck pain show less differential pain-expectancy judgments due to heightened pain-expectancy judgments for

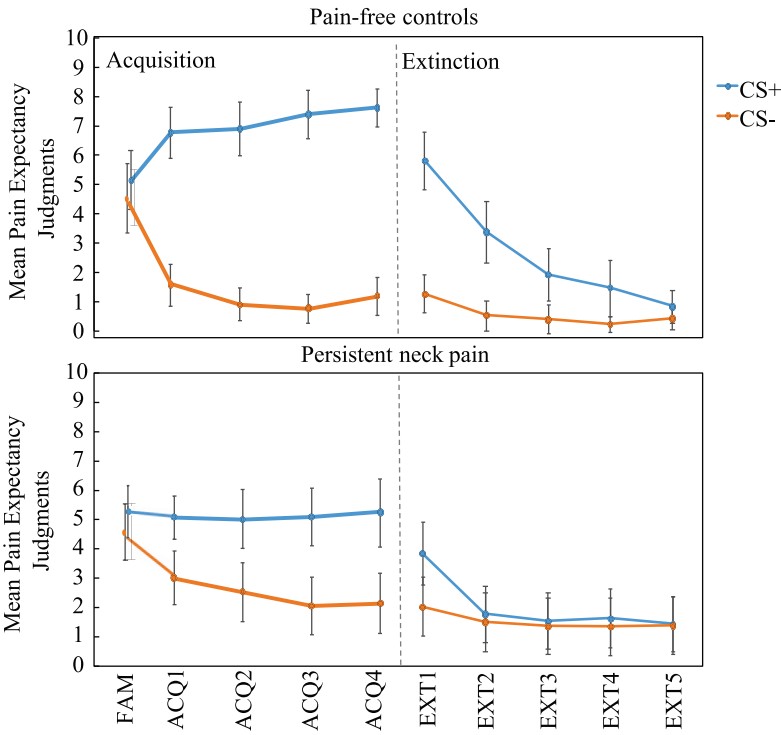

**Figure 3** Mean pain-expectancy judgments for the CS+ and CS− both for the pain-free control group (*n* = 30) and persistent neck pain group (*n* = 30) during the familiarization phase (FAM), the four blocks of acquisition (ACQ1-4), and the five blocks of extinction (EXT1-5), separately per block. Vertical bars represent 95% confidence intervals. CS+, conditioned stimulus paired with "PAIN" in 75% of the trials; CS−, conditioned stimulus always paired with "NO PAIN".

safety cues. In addition, and unexpectedly, lower pain-expectancy judgments for danger cues also contribute to the reduced differential predictive learning.

*Hypothesis 2: Do people with persistent neck pain show overgeneralization of pain-expectancy judgments compared to pain-free controls?*

We tested the generalization of pain-expectancy to novel but similar GSs and found a significant main effect of Stimulus Type, $F_{(2.82, 163.32)} = 50.22$, $p < 0.001$, $\eta_p^2 = 0.46$, and a significant interaction effect of Stimulus Type × Group, $F_{(2.82, 163)} = 7.53$, $p < 0.001$, $\eta_p^2 = 0.12$, suggesting group differences in pain-expectancy judgments (see Fig. 4). To probe group differences across the generalization phase, we first compared generalization gradients among groups, using linear trends generated by including CS+, GS1-5 and CS− expectancy judgments. The analyses revealed that patients display flatter generalization gradients compared with controls, $t_{(58)} = 4.07$, $p < 0.001$, Cohen's $d = 1.05$, $M_{patients}$ (SD) = −0.31 (0.72) vs. $M_{controls}$ (SD) = −1.12 (0.82).

Finally, CS+ and CS− expectancy ratings were compared to all other expectancy ratings within each group. Holm–Bonferroni corrected comparisons showed that CS+ expectancy ratings were significantly greater than all other stimuli within both groups (all ps < 0.015), showing that both groups had significantly high expectancy during the

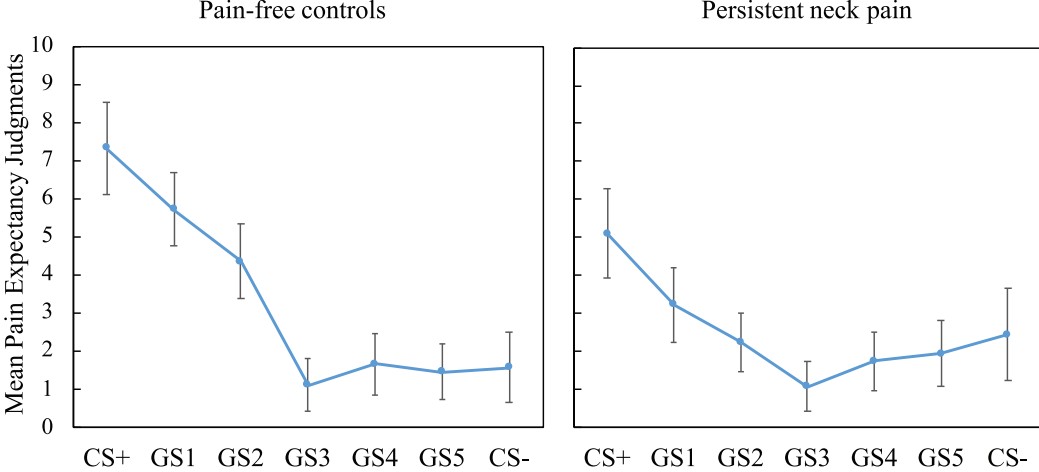

**Figure 4** Mean pain-expectancy judgments for CS+, GS1-5, CS− for the patient group (*n* = 30) as well as the pain-free controls (*n* = 30) during the generalization phase, averaged over blocks. Vertical bars represent 95% confidence intervals. CS+, conditioned stimulus paired with "PAIN" in 75% of the trials; CS−, conditioned stimulus always paired with "NO PAIN"; GS1, Generalization Stimulus most closely resembling the CS+.

CS+ relative to all other stimuli, this is generally referred to as a generalization decrement. For the contrasts relative to the CS−, only the CS+ expectancy was different ($p = 0.006$) in the patient group (all other ps > 0.12). In the controls, however, the GS1 and GS2 expectancy ratings were also significantly different to the CS− (ps < 0.001), suggesting that pain-expectancy ratings elicited by these stimuli were higher than for the safe neck position (CS−). This finding further reflects non-differential responding in the patient group.

*Hypothesis 3: Do people with persistent neck pain show resistance to extinction of pain-expectancy judgments compared to pain-free controls?*

The analysis of the pain-expectancy judgments during extinction yielded a main effect of Stimulus Type, $F_{(1, 58)} = 37.09$, $p < 0.001$, $\eta_p^2 = 0.34$, and a significant Stimulus Type x Block interaction, $F_{(4, 232)} = 30.25$, $p < 0.001$, $\eta_p^2 = 0.32$, likely reflecting diminishing CS+ expectancies, as compared to the CS−, across extinction blocks relative to the end of acquisition. In addition, we observed a significant Stimulus Type × Group interaction, $F_{(1, 58)} = 13.69$, $p < 0.001$, $\eta_p^2 = 0.13$, and a significant Stimulus Type × Block × Group interaction, $F_{(4, 232)} = 6.09$, $p = 0.01$, $\eta_p^2 = 0.06$. Visual inspection of the data (Fig. 3) indicated that the interaction might be driven by seemingly quicker extinction of (the already lesser) differential expectancies in the patient group. While no group differences during Extinction block 1, a Stimulus Type × Group RM ANOVA within Extinction block 2 was consistent with group differences $F_{(1, 58)} = 2.29$, $p < 0.001$, $\eta_p^2 = 0.26$. Inspection of Fig. 3 suggests that this effect was due to extinction of differential expectancies in the patient group. In the final extinction block, neither groups retained differential (CS+ vs. CS−) pain-expectancies, Controls: $t_{(29)} = 1.65$, $p = 0.11$; Patients: $t_{(29)} = 0.39$, $p = 0.70$, with no group differences in CS+ ratings, $t_{(58)} = -1.03$, $p = 0.31$) or CS− ratings, $t_{(58)} = -1.76$, $p = 0.08$.

## DISCUSSION

In this study we investigated whether people with persistent neck pain show deficits in predictive learning relative to pain-free controls. We hypothesized that people with persistent neck pain would show (1) less differential pain-expectancy learning, (2) overgeneralization of pain-expectancies to novel but similar neck postures and (3) reduced extinction of differential pain-expectancies. In line with our first hypothesis, we observed less differential pain-expectancies for the CS+ and the CS− for the patients as compared to controls. These results are broadly consistent with earlier findings on patients with chronic unilateral hand pain (*Meulders et al., 2014*), fibromyalgia (*Meulders et al., 2017*; *Meulders, Jans & Vlaeyen, 2015*) and chronic back pain (*Klinger et al., 2010*; Schneider et al., 2004). This finding appeared to be both due to reduced CS+ expectancy, and greater CS− expectancy in the neck pain group. With regards to our second hypothesis that people with neck pain would overgeneralize pain-expectancies relative to pain-free controls, results were partially aligned with our hypothesis. Here, both groups showed greater CS+ expectancy relative to all other stimuli, reflecting a common generalization decrement. However, when considering CS− expectancy, patients did not appear to discriminate it from any of the GSs, whereas controls did. This reduced discrimination among cues in the generalization phase was also consistent with the flatter generalization gradient in the patient group, providing supporting evidence for group differences in generalization. The possibility that people with neck pain tend to overgeneralize pain-expectancy is in line with existing literature on pain-expectancy judgments in people with persistent pain, including fibromyalgia and chronic hand pain patients (*Meulders et al., 2014*, *2017*). It is also interesting to note that in general responses to both the CSs and the GSs were lower for patients than in the healthy control group. Such blunted fear responding and wider fear generalization gradients have been recently observed in people who experienced childhood maltreatment (*Lonsdorf, 2019*). Because differences in differential learning can influence generalization, we explored whether acquisition phase differential learning mediated generalization linear slopes (see Supplemental File for full details). Direct and indirect (mediated) effects of group were of approximately equal weight, suggesting that as least part of the generalization effects was independent of acquisition learning. Contrary to our third hypothesis—that people with neck pain would show delayed extinction of differential learning—differential responding appeared to disappear quicker in people with neck pain. The rate of extinction learning however is likely not different from healthy controls, patients simply seem to extinguish quicker because they start off with less differential responding and overall lower responses at the end of acquisition. It is notable that although delayed extinction is commonly identified as a possible impairment contributing to the maintenance of chronic pain, there is currently little supporting evidence for the thesis (*Harvie et al., 2017*; also see Schneider et al., 2004).

### Differential learning

The most robust finding in this study was the reduced differential learning in people with persistent neck pain relative to pain-free controls. This reduced differential learning was

driven by both lower pain-expectancies for the CS+, and higher pain-expectancy for the CS-. In prior studies, in people with persistent pain and people with anxiety disorders (*Harvie et al., 2017*; *Lissek et al., 2005*), reduced differential learning is typically the primary result of heightened CS− expectancies—referred to as impaired safety learning. The current study shows preliminary evidence that safety learning deficits are present in some individuals with neck pain. The deficits in differential learning indicate a tendency towards reduced efficiency in selectively evaluating safety cues. That is, patients in general may be able to learn when to expect pain, and when not too, but with greater uncertainty. With regards to the lower CS+ pain-expectancy judgements, a similar data pattern (reduced responding to CS+ in patients compared to healthy controls, albeit not significant) was observed previously (*Meulders et al., 2014*), we speculate that expectancy judgements may be contaminated by certainty judgements. That is, one may have a high degree of pain-expectancy, but a low degree of certainty. For example, one might expect rain, but lack certainty regarding that judgment. This example shows that expectancy and certainty do not necessarily converge. This lack of certainty would result in people tempering their CS+ expectancy ratings relative to if they had both a high degree of expectancy and a high degree of certainty. Greater overall uncertainty in patients may arise from factors such as reduced self-efficacy/ self-confidence in addition to impaired predictive learning.

## Limitations and future directions

The results of this study are subject to several limitations. The cross-sectional nature of the design limits our capacity to reveal causal relationships between the chronic pain status and altered expectancy learning outcomes. It is thereby unclear whether these deficits contribute to chronic pain, or are a result of it. Indeed, prospective studies are required in this endeavor. Notably, the greatest deficits may occur in the most severely afflicted patients. The patients in the current study in general had only mild-moderate severity with respect to pain and disability. This may in part be due to the exclusion of subjects on opioid medications. While this may have excluded more severe patients, it strengthened the study in preventing the confounding effect of opioids on cognitive performance. Further, participants showed no, or low levels, of co-morbid psychological dysfunction, which may be critical drivers of the constructs under investigation. Crucially, this is different to other studies where participants did show significant group differences in psychological variables (*Meulders et al., 2014*, *2017*; *Meulders, Jans & Vlaeyen, 2015*). That we nonetheless found group differences in predictive learning, despite less disability and psychological dysfunction compared to previously studied patient groups, is remarkable and raises the possibility that learning deficits may exist independent of psychological factors and may be present even in less severe cases of persistent pain. Finally, as we have disproportionately pioneered this line of research, replication from independent labs is needed, and indeed beginning to emerge (*Both et al., 2017*; *Heathcote et al., 2020*).

## CONCLUSIONS

We aimed to determine whether people with chronic neck pain show deficits in expectancy learning relative to pain-free controls. We found evidence for our first hypothesis

observing less differential pain-expectancies for the people with persistent neck pain as compared to pain-free controls, suggesting reduced efficacy in identifying predictors of pain. Some preliminary support was also found for overgeneralization of pain-expectancies relative to controls, in that generalization gradients were flatter in the patient group. Contrary to our final hypotheses, people with neck pain did not show reduced extinction learning. These findings may be relevant to understanding behavioral aspects of chronic pain, however prospective studies are needed.

### Funding

Daniel Harvie is supported by an Early Career Research Fellowship from the National Health and Medical Research Council of Australia. Ann Meulders was supported by a Vidi grant from the Netherlands Organization for Scientific Research (NWO), The Netherlands (grant ID 452-17-002). Ann Meulders is also supported by a Senior Research Fellowship of the Research Foundation Flanders (FWO-Vlaanderen), Belgium (grant ID: 12E3717N). The funders had no role in study design, data collection and analysis, decision to publish, or preparation of the manuscript.

### Grant Disclosures

The following grant information was disclosed by the authors:
National Health and Medical Research Council of Australia.
Vidi grant from the Netherlands Organization for Scientific Research (NWO),
The Netherlands: 452-17-002.
Senior Research Fellowship of the Research Foundation Flanders (FWO-Vlaanderen),
Belgium: 12E3717N.

### Competing Interests

The authors declare that they have no competing interests.

### Author Contributions

- Daniel S. Harvie conceived and designed the experiments, analyzed the data, prepared figures and/or tables, authored or reviewed drafts of the paper, and approved the final draft.
- Jeroen D. Weermeijer conceived and designed the experiments, performed the experiments, analyzed the data, prepared figures and/or tables, authored or reviewed drafts of the paper, and approved the final draft.
- Nick A. Olthof conceived and designed the experiments, performed the experiments, authored or reviewed drafts of the paper, and approved the final draft.
- Ann Meulders conceived and designed the experiments, analyzed the data, prepared figures and/or tables, authored or reviewed drafts of the paper, and approved the final draft.

## Human Ethics

The following information was supplied relating to ethical approvals (i.e., approving body and any reference numbers):

The Griffith University Human Research Ethics Committee granted Ethical approval (2017/710).

## Data Availability

The raw data set is available in the Supplemental Files.

## Supplemental Information

Supplemental information for this article can be found online at http://dx.doi.org/10.7717/peerj.9345#supplemental-information.

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
