# Peer review of "Learning to predict pain: differences in people with persistent neck pain and pain-free controls"

_PeerJ, doi:10.7717/peerj.9345_

## Round 0.1 · original submission · Minor Revisions

I have now received three reviews from experts on the fields on your manuscript. The evaluations of the reviewers are remarkably consistent: they all propose a "minor revision" verdict, and I agree with their assessment. All the reviews are constructive and clear. I invite you to address the reviewers' concerns, focusing in particular on the hypothesis, the method and analyses (e.g. sample size, checking normality for Anovas) and the interpretation of the data (e.g. of pain expectancy judgments). I also invite you to avoid using SPSS documents when you upload raw data.

Reviewer 1 ·

Basic reporting

Overall the article is well written
To comply with the study being self contained the justification for the sample size should be stated in this paper and not referenced to a previous paper (line 126)
The raw data is contained, however this would be best done as an excel document for greater access instead of an SPSS document.

Experimental design

No Comment

Validity of the findings

A limitation that should be included or a future options for research is for the results to be verified by another research lab. All the findings to date in this area have been done by one lab and need to be independently replicated by another lab. This should be mentioned in the discussion.

Additional comments

1. In table 1 please include the mean difference scores and a 95%CIs
2. For the analysis plan. Please include a small section for each hypothesis as to how the planned analysis would test the hypotheses the benchmark you used for determining if the statistical test confirmed or rejected the hypothesis. For example with the first hypothesis investigating differential pain expectancy judgements explain how the RM ANOVA would determine if there was a difference. This may be more easily down in the previous section when describing the experiment and explaining how the different components of the experiment were used to investigate the different hypothesis.
3. At the moment there are many statistical outcome measures i.e. corrected p values and partial eta squared. Why were both used and what was used to determine if the hypothesis was met. How were the eta values interpreted?
4. Line 272 states there was no significant conflict in the results, indicating that there may have been some conflict. Please expand on what the differences were and which analysis software was used for which part of the analysis or if the different software was used by different authors.
5. Throughout the statistical analysis plan the term “planned” is used frequently. Is there any available evidence, such as a published analysis plan, that these were in fact “planned”. If no such evidence is available then please state this explicitly.
6. If the authors could review the introduction to ensure that the key themes are explained in the most clear way possible. My specific suggestion here is to ensure that there is a clear link as to how the experiment answers the research question- this may be better done in the methods section. I feel this will improve the readability of the study for people unfamiliar with the techniques used. The risk is that if the reader is not able to understand how the experiment answers the research question then they are more likely to question the validity of the findings. I feel that if point 2 is adequately addressed this may resolve the current point.
Minor Comments
7. It is concerning how many times the term “may” is used in the introduction. As the conclusion rightly points out longitudinal studies are required to determine how much associate learning is important in the development of chronic neck pain. The consistent use of the term “may” implies that there are still many “leaps” in the model. I do not think this should be changed but just an observation.

Reviewer 2 ·

Basic reporting

no comment

Experimental design

no comment

Validity of the findings

no comment

Additional comments

This study compared the predictive learning between patients with chronic neck pain and pain-free controls, and three hypotheses were tested. Data in the acquisition stage showed significantly reduced differential pain-expectancy learning in patients than controls, as evidenced by the higher pain judgments for CS-, and lower pain judgments for CS+ in the acquisition stage. Data in the generalization stage showed flatter generalization gradients in the patient group than control group. However, data in the extinction stage did not show significant between-group difference. The authors interpreted this pattern of data as deficits in predictive learning in patients with neck pain. Overall, it is an interesting study, and I like this study that is well motivated and well written. The experimental design and data analysis is rational. I have several suggestions that the authors can consider for improvement.

For the hypothesis that patients have less differential acquisition in pain expectancy judgments than controls, the reduced differential learning was driven by lower expectancies for CS+ and higher expectancies for CS-. While the higher expectancies for CS- could be reflecting the reduced efficiency in evaluating safety cues, why the patients have lower expectancies for CS+? Can the authors more explicitly discuss this pattern of finding?

Have the authors have considered the possibility that the observed flatter generalization gradient in the patient group could be resulting from their less efficient learning in the acquisition stage. Maybe a mediation analysis can help clarify this problem, with X for group (patients vs. controls), M for acquisition learning (e.g., differential learning), and Y for generalization gradient. At least, the possible relationship should be discussed in the manuscript.

I would suggest the authors test whether the inter-individual variability in predictive learning deficit relates to pain severity/disability across the patients. This can help clarify the relationship between pain and predictive learning.

·

Basic reporting

The English is clear and professional,
The introduction set the work in context.
The figures were clear and readable, and helped to illustrate the key findings.
The structure conformed to Peer J policies.
The raw data was supplied but I could not open the file to check it.

Experimental design

This study took a novel approach using sound experimental methods to the question of how fear avoidance type behaviours impacts the way individuals with chronic pain evaluate perceived risk. This issue is that failure to differentiate between cues that truly predict danger and those that do not, can lead to inappropriate fear which exacerbates fear-avoidance behaviours, making this an important study.

The methods were clear and well-described.

The only weakness is that the assumptions of normality for the ANOVAs were not described nor were this results reported. This needs to be addressed.

Validity of the findings

The conclusions were clear and well stated. The authors were clear in restating each hypothesis and the relevant findings related to that hypothesis.

The most robust and novel finding was that deficits in differential learning which suggest reduced efficiency in selectively evaluating safety cues in the chronic neck pain patients, ie. patients were able to learn when to expect pain, and when not too, but with greater uncertainty than healthy controls.

This is an important finding as it suggests altered discrimination across the spectrum which will fuel future studies in the area.

It was important that in the limitations, the authors mentioned that the study was cross sectional, and that patients in the study only had mild to moderate pain, likely because opioid-use was an exclusion criteria

---

## Round 0.2 · accepted · Accept

I am glad to tell you that your paper has been accepted for publication.